# Effects of Sarcosine (N-methylglycine) on NMDA (N-methyl-D-aspartate) Receptor Hypofunction Induced by MK801: In Vivo Calcium Imaging in the CA1 Region of the Dorsal Hippocampus

**DOI:** 10.3390/brainsci14111150

**Published:** 2024-11-16

**Authors:** Yi-Tse Hsiao, Ching-Yuan Chang, Ting-Yen Lee, Wan-Ting Liao, Wen-Sung Lai, Fang-Chia Chang

**Affiliations:** 1Department of Veterinary Medicine, School of Veterinary Medicine, National Taiwan University, Taipei 10617, Taiwan; ythsiao@ntu.edu.tw (Y.-T.H.); r08629017@ntu.edu.tw (C.-Y.C.); r08629009@ntu.edu.tw (T.-Y.L.); liaoteresa8431@ibms.sinica.edu.tw (W.-T.L.); 2Neurobiology and Cognitive Science Center, National Taiwan University, Taipei 10617, Taiwan; wslai@ntu.edu.tw; 3Department of Psychology, National Taiwan University, Taipei 10617, Taiwan; 4Graduate Institute of Brain and Mind Sciences, National Taiwan University, Taipei 10617, Taiwan; 5Graduate Institute of Acupuncture Science, College of Chinese Medicine, China Medical University, Taichung City 40402, Taiwan; 6Department of Medicine, College of Medicine, China Medical University, Taichung City 404333, Taiwan

**Keywords:** glycine, Miniscope, calcium imaging, schizophrenia, hypofunction model

## Abstract

Background: Hypofunction of the glutamate system in the brain is one of the pathophysiological hypotheses for schizophrenia. Accumulating animal and clinical studies show that sarcosine (N-methylglycine), a glycine transporter-1 inhibitor, is effective in ameliorating the negative and cognitive symptoms of schizophrenia. The aims of the present study were to observe the effects of sarcosine on neuronal activity in the dorsal CA1 (dCA1) hippocampal neurons within an NMDA receptor hypofunction model induced by MK801. Methods: We applied in vivo calcium imaging to observe the dynamics of fluorescence from the dCA1 hippocampal neurons when the mice were exploring in an open field. Using this tool, we directly measured and compared neuronal properties between sarcosine-treated and untreated mice. At the same time, the physiological function of the neurons was also quantified by measuring their place fields. Results: Our data demonstrated that MK-801 (0.2 mg/kg) diminished the fluorescence intensity of dCA1 neurons that had been genetically modified with a calcium indicator. MK-801 also significantly increased the correlation coefficient between the fluorescence dynamics of pairs of cells, a feature that may be linked to the symptom of disorganization in human patients with schizophrenia. The spatial correlations of place fields in the mice were impaired by MK-801 as well. Injected sarcosine (500 mg or 1000 mg/kg) significantly alleviated the abovementioned abnormalities. Conclusions: Our data provide evidence to support the use of sarcosine to alleviate symptoms of schizophrenia, especially hippocampus-related functions.

## 1. Introduction

Hyperactivity of the dopamine system and hypofunction of the glutamate system are two leading theories for the pathophysiology of schizophrenia [1,2,3,4]. The symptoms of schizophrenia have been categorized into positive and negative symptoms [5], which may relate differently to these theories. For instance, N-methyl-D-aspartate (NMDA) receptor hypofunction has been hypothesized to be a key cause of negative symptoms of schizophrenia [2,3,4,6,7]. Cognitive impairment is also a serious problem in patients with schizophrenia [6,7,8]. Drugs that have been developed to potentiate the function of NMDA receptors have been found to be effective in improving cognitive deficits in schizophrenia [6,7,9,10]. Although the NMDA receptor is critical, the pharmacological mechanisms for currently available antipsychotics are mainly focused on the dopamine receptor system [11]. Although dopamine-targeting drugs are effective at alleviating positive symptoms, they have limited efficacy in treating negative symptoms; therefore, developing glutamate-targeting drugs has gained much attention in recent years [6,9].

One of the straightforward strategies for enhancing the function of the NMDA receptor is the administration of glutamate. Unfortunately, this therapeutic strategy is limited by the blood–brain barrier [12], which restricts peripheral glutamate from passing to the central nervous system. Alternately, the modulation sites on the NMDA receptor become a main target for developing antipsychotics for schizophrenia. Activation of the NMDA receptor requires not only glutamate binding but also synaptic depolarization and activation of glycine modulatory sites [6]. Therefore, the glycine modulatory site on the NR1 subunit of the NMDA receptor is a potential therapeutic target. In clinical tests, oral administration of glycine significantly improves negative syndromes [13,14]. However, the effective dose needed to cross the blood–brain barrier is high (0.8 g/kg/day) [13,14] and may cause upper gastrointestinal tract discomfort [14]. In recent years, another amino acid, sarcosine (N-methylglycine), was found to be effective as both add-on therapy and monotherapy for schizophrenia in clinical trials [15,16,17,18]. The recommended dose for treating schizophrenia is 2 g/day [15], which is far less than the recommended dose of glycine. Sarcosine was first recognized as an adjuvant medication for schizophrenia because sarcosine indirectly increases synaptic glycine levels by selective inhibition of glycine transporter 1 [19]. The action of sarcosine seems to have multiple underlying pharmacological mechanisms. Pei et al. demonstrated that sarcosine also increases D-serine, a glycine modulatory site agonist in CSF [20]. Moreover, they also reported that sarcosine decreases the surface trafficking of the NR1 subunit of the NMDA receptor and may coactivate directly on the glycine modulatory site [20]. In their experimental design, they used several animal models, such as transgenic mice, MK-801-exposed mice [20,21,22], and brain slices of wild-type mice, to test the underlying therapeutic mechanisms of sarcosine. We noticed some interesting data from their results. First, PET/CT showed that sarcosine ameliorated the MK-801-induced high standardized uptake value of ^18^F-FDG in the dorsal hippocampus. Second, sarcosine enhances the NMDA receptor-mediated field excitatory postsynaptic potential slope in wild-type mouse hippocampal slices. Pei et al. also performed several behavioral tasks to test whether sarcosine improves MK-801-induced behavioral deficits. In their experiments, they used several groups of animals for various approaches to portray the effect of sarcosine on schizophrenia. They demonstrate that intraperitoneal injection of MK-801 causes a decrease in exploratory behavior, sensorimotor gating function, learning and memory functions, and an increase in depressive-like behavior in mice, which are schizophrenia-related behaviors [20]. In addition, sarcosine alleviated the abovementioned schizophrenia-related behavior in an MK-801 mouse model of schizophrenia [20]. We assumed that directly observing the influences of sarcosine on hippocampal neurons in their MK-801-induced schizophrenia-like model as the animals are freely behaving may provide evidence for explaining the potential impacts of sarcosine on improving neuronal functions that are processed in the hippocampus. We used in vivo calcium imaging in freely moving mice to observe their dorsal hippocampal CA1 (dCA1) neurons [23]. We acquired over 400 neurons in three mice (Table 1) and analyzed their properties and physiological functions. The best-known physiological function of dorsal hippocampal cells is their ability to encode spatial memory. Neurons in the hippocampus that preferentially fire when an animal visits a specific place are referred to as place cells [24]. The aim of the present study was to directly observe neurons in the dorsal hippocampus of the MK-801-induced schizophrenia-like model [20,21,22] and test whether sarcosine rescues the abnormality of neurons in terms of cell properties. We compared the properties of single, paired, and assembled cells between sarcosine-treated and untreated neurons. Although glutamatergic transmission is implicated in schizophrenia and multiple brain regions are involved [20], we narrowed down the target research region to dCA1 as a previous study demonstrated that MK-801 induces abnormalities in the dorsal hippocampus, which are improved by sarcosine [20]. Our results may pave the way for observing how the dCA1 neurons are influenced in MK-801-induced schizophrenia-like models and the ability of sarcosine to alleviate the abnormalities.

## 2. Materials and Methods

### 2.1. Substances and Animals

Sarcosine (#157741000; Acros Organics; Fair Lawn, NJ, USA) or MK-801 (#M107; Sigma; Merck, Darmstadt, Germany) were separately dissolved in PBS and used immediately when performing the experiment. Concentrations of 0.02 mg/mL for MK-801 and 100 mg/mL or 50 mg/mL for sarcosine were prepared in PBS, which resulted in 0.2 mg/kg of MK801 and 1000 mg/kg or 500 mg/kg of sarcosine in an injection volume of 0.3 mL for a 30 g mouse [20]. Three male C57BL/6JNarl (26–30 g; National Laboratory Animal Center, Taipei, Taiwan) mice were used. The mice were housed in a temperature-controlled room (23 ± 1 °C) with a 12:12 h light: dark cycle. Food and water were available ad libitum. Because we planned to observe the putative place cells in the dCA1, the mice were pretrained for exploration in an open field. The animals were allowed to explore freely in a 15 cm × 30 cm open field for 10 min for at least 3 days before the surgery. All procedures performed in this study were approved by the National Taiwan University Animal Care and Use Committee (Approval No.: NTU-109-EL-00029).

### 2.2. Surgery and Viral Injection

The surgery and viral injection protocols were adapted from Resendez et al.’s report [23] and the UCLA Miniscope Team’s surgery guidelines (http://miniscope.org/index.php/Surgery_Protocol (accessed on 5 June 2018)). A video demonstrating the protocol can be found in our previous article [25]. The mice were subcutaneously injected with an analgesic (buprenorphine, 0.03 mg/kg) and atropine (0.04 mg/kg) to prevent the accumulation of saliva and then anesthetized with isoflurane (induction, 5%; maintenance, 1–2%) in oxygen. Two stainless steel screws were surgically anchored onto the left frontal and interparietal bones. A craniotomy with a diameter of 2.0 mm was performed with the center at AP −2.25 mm, ML +1.8 mm relative to bregma. The dura and cortex were carefully removed by suction with a 27 G needle until the corpus callosum above the dCA1 was exposed. Five hundred nanoliters of viral vectors (AAV-syn-jGCaMP7s-WPRE; 1.5 × 10^13^, # 104487-AAV9; Addgene viral prep; Watertown, MA, USA) were injected into the dCA1 (AP: −2.0 mm, ML: +1.5 mm, DV: 1.5 mm, relative to bregma). After the injection had infused for 10 min, a gradient index (GRIN) lens (# 64519, diameter 1.8 mm, length 4.31 mm; Edmund Optics; Barrington, NJ, USA) was implanted onto the dCA1 (AP: −2.25 mm, ML: +1.8 mm, DV: 1.3 mm, relative to bregma) and glued with little light cure adhesive (#3321; Loctite; Düsseldorf, Germany). The anchor screws and GRIN lens were then fixed with dental cement (Tempron; Tokyo, Japan). Since the GRIN lens needs to protrude from the surface of the skull as much as possible and may be easily damaged by the animal, molding silicone rubber (ZA22 Thixo; Zhermack; Badia Polesine, Italy) was applied to the lens and then covered with a thin layer of dental cement to protect the GRIN lens. After 6 weeks of incubation, the mice were sedated with isoflurane (induction, 5%; maintenance, 0.8–1.5%) and fixed in a stereotactic frame. Their silicone rubber was removed, and the surface of the GRIN lens was cleaned with 75% alcohol. A custom-made Miniscope with a baseplate (V3; http://miniscope.org (accessed on 5 June 2018)) was placed onto the lens under to monitor the data acquisition software (http://miniscope.org/index.php/Data_Acquisition_Software (accessed on 5 June 2018)). The height of the baseplate was adjusted with the stereotactic frame until we saw a bright puncta, which represented jGCaMP7s-expressing neurons. The baseplate was carefully cemented in its present location and detached from the Miniscope. After the dental cement was completely dry, the mice were moved to a box for recovery and then placed back in their home cage. The location of implantation was confirmed by histology slices after finishing the whole experiment (Figure 1A).

### 2.3. Experimental Procedures

The mice were habituated to the procedure of mounting the Miniscope onto the baseplate on a daily basis for 3 days; then, the animals carried the Miniscope and explored freely in a 15 cm × 30 cm open field for 4 min for training purposes. After the mice were habituated to the handling procedures and performed well in the open field (i.e., the walking path covered most of the open field), we started to execute the experiments. Four sessions of drug testing experiments were randomly performed at 48 h intervals to wash out the drug. Each session contained two 4 min trials (trials A and B) with a drug administration period in between (Figure 1B). In each trial, the mice were also allowed to move freely in the same 15 cm × 30 cm open field in the same environment on training days. The raw calcium imaging videos were acquired by the Miniscope UCLA data acquisition software. At the same time, the behavioral videos were recorded and analyzed by EthoVision tracking software (EthoVision XT 14; Wageningen, the Netherlands). After finishing trial A, the mice were intraperitoneally injected with PBS (0.01 mL/g) or MK-801 (0.02 mg/kg) and waited 5 min and then injected with PBS (0.01 mL/g) or sarcosine (500 mg/kg; SAR500) or sarcosine (1000 mg/kg; SAR1000), which created a combination of 1. PBS + PBS (vehicle control), 2. MK-801 + PBS (untreated session), 3. MK-801 + SAR500 (low dose treatment), and 4. MK-801 + SAR1000 (high dose treatment). Then, the mice waited in their home cage for 20 min prior to returning to the open field for experiment trial B.

### 2.4. Data Analysis

#### 2.4.1. Processing of the Ca^2+^ Imaging Videos

The calcium imaging videos were processed with MIN1PIPE [26] (Figure 1C,D). MIN1PIPE removed the background noise and corrected the motion artifacts of the videos. The seeds were first automatically selected by MIN1PIPE and then inspected by investigators to exclude the double-selected seeds. We did not compare the fluorescence dynamics of the same cells across days; therefore, the assigned seed numbers were different across sessions (e.g., the assigned cell numbers in the control session are different from those in the MK-801 session). The processed fluorescence intensity traces of each cell were then analyzed by custom written programs in MATLAB (R2016b; MathWorks; Natick, MA, USA) described as follows.

#### 2.4.2. Comparisons of Grouped Fluorescence Intensity

Miniscope calcium imaging enables us to observe individual neuronal activity. However, slight shifts in neuron positioning across days make tracking the same neurons challenging. To evaluate the effects of sarcosine on restoring network-level neuronal activity while avoiding the complications of tracking identical neurons across days, we analyzed randomly selected assembled neuronal responses rather than focusing on the exact same individual neurons. The fluorescence intensity traces of 20 cells were randomly selected, and their intensity ratio between trial B and trial A was calculated and averaged, which resulted in a sampling unit of 20 neurons, given the finding by Ghandour et al. that approximately 10% of neurons (~20 neurons in the present study) in the dCA1 have ensemble activity [27]. This procedure of random selection and averaging was performed 1000 times (MATLAB), which gave 1000 intensity ratios (Figure 2A–C). To ensure this number was not too arbitrary, we also sampled 30 (Figure 2D–F) and 50 (Figure 2G–I) neurons as additional comparisons. This iterative sampling, similar to bootstrapping, provides a stable estimate of neural network responses. The 1000 iterations follow common bootstrapping practices and typically yield robust statistical reliability [28]. This method thus allowed us to assess sarcosine treatment effects on network-level activity under MK801 conditions.

#### 2.4.3. Correlation Matrix

To understand the effect on cell–cell interactions, we created a correlation matrix to observe the fluorescence correlations between pairs of dCA1 neurons (Figure 3). The fluorescence intensity traces that proceeded by MIN1PIPE were paired and computed by the MATLAB function corrcoef.m, which calculates the Pearson correlation coefficients of two traces.

#### 2.4.4. Spatial Correlation

The physiological functions of dCA1 cells were tested by measuring the alterations of place fields and comparing the field of trial A with trial B. In detail, we first binned the rectangle open field into a 1 cm resolution map and matched the time of fluorescence intensity traces to the open field bins where the mice explored. The maximum fluorescence intensities of each bin are represented as the fluorescence map (Figure 4A). The fluorescence maps were then averaged across the width of the open field, which converted the fluorescence map matrixes into the length of fluorescence map vectors. We then calculated the Spearman correlations between the fluorescence map vectors of trial A and trial B (Figure 4B).

### 2.5. Statistics

To assess the significance of differences between the four treatment groups (control, MK-801, MK-801 + SAR500, and MK-801 + SAR1000), a non-parametric bootstrapping approach was employed (MATLAB). For each group, the data was resampled with replacement 1000 times to generate a distribution of bootstrapped means. Refer to Table 1 for the number of samples in each resampling that matched the original sample size for each group. After generating the bootstrapped distributions, a one-way analysis of variance (ANOVA) was performed on the bootstrapped means to evaluate overall differences between the treatment groups. When the ANOVA indicated a significant effect (*p* < 0.05), post-hoc pairwise comparisons were conducted using Tukey’s honestly significant difference (HSD) test to identify which specific groups differed. To assess the significance of differences in the cumulative distributions, we applied the Kolmogorov–Smirnov (K-S) test. Significance analyses were computed by built-in tools in MATLAB and tested with SPSS (Version: 10.0.7, IBM, New York, NY, USA).

## 3. Results

### 3.1. Single Neuron Fluorescence Intensity

The activities of the right dCA1 neurons of three mice are represented by calcium indicators. Figure 1A,C demonstrate that jGCamp7s expressed in the dCA1 and GRIN lens are located above the pyramidal layer. Their dynamic changes were videotaped by a Miniscope mounted on the subject’s head (Appendix A). The fluorescence dynamics of each seed were picked out and processed, including movement correction (MC), background subtraction, and noise removal, and represented by fluorescence intensity traces (Figure 1D, Appendix A). By observing the videos (Appendix A), we noticed that intraperitoneal (IP) administration of 0.2 mg/kg MK-801 seemed to decrease the fluorescence activities of neurons. Therefore, we further displayed their fluorescence by gradient gray lines for better visualization (Figure 1E–H and Appendix A). The time intervals between sessions were more than two days for washing out the previous drugs. Because we did not aim to analyze the alterations of the same cell across sessions, the seed numbers and the number of detected neurons were different from session to session but identical within trials (Table 1). The alterations of fluorescence intensities were calculated in Figure 1I, which measured the mean delta fluorescence intensities of each trace between trial A and trial B (Mean ± SEM). Although the fluorescence intensities in trial B were usually lower than those in trial A, which resulted in negative mean values (Figure 1I), MK-801 still significantly reduced the brightness of fluorescence when compared to the delta fluorescence intensities of the vehicle control (Figure 1I, PBS + PBS vs. MK-801 + PBS Bootstrapping and Tukey’s post hoc comparison, *p* < 0.01). Supporting the observation in Figure 1G,H, sarcosine ameliorated the MK-801-induced reduction. Moreover, SAR500 (sarcosine 500 mg/kg) showed better restoration than SAR1000 (1000 mg/kg) (Figure 1I, MK-801 + SAR500 vs. MK-801 + PBS Bootstrapping and Tukey’s post hoc comparison, *p* < 0.01; MK-801 + SAR1000 vs. MK-801 + PBS, *p* < 0.01). Fluorescence intensities from the other mice (mouse #1; Appendix A and mouse #2; Appendix A) also demonstrated better restoration in SAR500 than SAR1000. Although in Figure 1I, alterations in fluorescence intensities were quantified with each individual neuron considered as a single statistical sample, we were curious about the fluorescence dynamics of cell ensembles. Therefore, we performed two kinds of analyses (Figure 2 and Figure 3) that took multiple cells or paired cells as a measurement.

### 3.2. Grouped Fluorescence Intensity

We randomly sampled 20 cells, averaged the fluorescence intensities of every mouse 1000 times, and plotted the cumulative distributions of their ratios between trials (trial B/trial A) (Figure 2A–C), given the finding by Ghandour et al. that approximately 10% of neurons (~20 neurons in the present study) in the dCA1 have ensemble activities, which hint that they are hippocampal memory engrams [27]. Figure 2A–C are the cumulative distributions of mice 1 to 3, respectively. Compared with the vehicle control sessions (Figure 2A–C; black lines), the distributions of the trial B/A ratios were shifted toward left after administrating MK-801 (gray lines), which suggests that MK-801 reduced the fluorescence of any 20 neurons (control vs. MK-801+PBS, K-S test; mouse #1, *p* < 0.01, test statistic = 0.12; mouse #2, *p* < 0.01, test statistic = 0.87; mouse #3, *p* < 0.01, test statistic = 0.18). Moreover, SAR500 (red lines) consistently improved the fluorescence intensity of all three mice since their cumulative distributions were shifted toward the right (MK-801 + PBS vs. MK-801 + SAC500, K-S test; mouse #1, *p* < 0.01, test statistic = 0.50; mouse #2, *p* < 0.01, test statistic = 0.89; mouse #3, *p* < 0.01, test statistic = 0.63). However, the effect on the fluorescence of SAR1000 (blue lines) was not very consistent across the three mice. Although SAR1000 significantly increased the intensity compared with MK-801 (MK-801 + PBS vs. MK-801 + SAC1000, K-S test; mouse #1, *p* < 0.01, test statistic = 0.38; mouse #2, *p* < 0.01, test statistic = 0.30; mouse #3, *p* < 0.01, test statistic = 0.49), a dramatic improvement after SAR1000 injection was evident only in mouse #3 (Figure 2C, blue line). We also analyzed the ratio distributions with sampling units of 30 (Figure 2D–F) and 50 (Figure 2G–I) neurons to test whether the sampling size might affect the distributions. Consistently, Figure 2D–I demonstrated a similar tendency of distribution to those of Figure 2A–C. Therefore, the data suggest that MK-801 may suppress the activity of dCA1 ensembles and that sarcosine 500 mg/kg can ameliorate the suppression.

### 3.3. Pairwise Fluorescence Correlation

Next, we were still interested in knowing the cell–cell activities of dCA1; therefore, we measured the correlation matrix between pairwise fluorescence traces (Figure 3 and Appendix A). In Figure 3, the correlation coefficients are variables for measuring the alterations of cell pairs. Interestingly, MK-801 significantly increased the correlation coefficients of pairwise fluorescence intensities (Figure 3B). The percentage of high-correlation cell pairs (R > 0.5) was 9.25% in the MK-801 session (Figure 3A) and 3.87% in the control session (Figure 3B). Both SAR500 and SAR1000 halted the increase in the percentage of high-correlation pairs (Figure 3C, SAR500, 4.34% and Figure 3D, SAR1000, 4.11%). For the statistical test, we compared the alterations of correlation coefficients between baseline (trial A) and postinjection (trial B). The data showed a significant increase in delta cell–cell correlations after the administration of MK-801 (Figure 3E) in comparison to those of PBS. SAR500 or SAR1000 reduced the increases from MK-801 to the level near vehicle control. Because the tendency of the correlation coefficient (elevated after MK-801 administration) is different from the tendency of fluorescence intensity (decreased after MK-801 administration), we compared the pairwise mean fluorescence intensities to the correlation coefficients in trial B of the MK-801 session and noticed a moderated negative correlation (R = −0.587, *p* < 0.01), which suggested that cells paired with low fluorescence intensities tend to show high cell–cell correlation.

### 3.4. Spatial Correlation of Fluorescence Map

Although the above data represented the properties of neurons in the dCA1 under MK-801 or sarcosine treatment, we were still curious about the physiological functions of these neurons. Most of the neurons in the hippocampus have a property of place preference, which means that the specific neurons are activated only when the animal is located in a specific area [24]. Therefore, we computed the fluorescence map (Figure 4A) of each cell and measured the spatial correlations between trials A and B. The spatial correlation significantly decreased to −0.094 ± 0.019 after the injection of MK-801 (Figure 4B, Bootstrapping and Tukey’s post hoc comparison, *p* < 0.01). Moreover, SAR500 or SAR1000 both reversed the decreases in spatial correlations induced by MK-801 (Figure 4B, post hoc comparison with MK-801 session, *p* < 0.01). SAR500 even improved the spatial correlation when compared with the control session (Figure 4B, post hoc comparison with the control session, *p* < 0.01). These data suggest that the physiological functions of dCA1 were impaired by MK-801 but rescued by sarcosine.

## 4. Discussion

The aims of the present study were to observe neuronal activity in the dCA1 region under conditions of NMDA receptor hypofunction induced by MK801, which has been used in previous studies to model certain aspects of schizophrenia-like phenotypes [20,21,22]. While this approach is widely recognized to simulate NMDA receptor dysfunction, a core feature implicated in schizophrenia pathophysiology [2,3,4,9], we acknowledge that the characterization of this animal model as fully representative of schizophrenia remains debated. Thus, we cautiously propose that this model can be considered an NMDA receptor hypofunction model, which may mimic some aspects of schizophrenia for specific experimental purposes, such as evaluating the effects of sarcosine on dCA1 neurons. We used in vivo calcium imaging to observe the fluorescence dynamics of dCA1 neurons and analyzed three aspects of influences, including fluorescence intensity (either single (Figure 1) or assembled cell (Figure 2)), cell–cell correlations (Figure 3), and cellular function (Figure 4). The data showed that MK-801 decreased fluorescence intensity, increased cell–cell correlation, and impaired the place maps of dCA1 neurons. However, sarcosine could ameliorate the abovementioned influences. Using freely behaving calcium imaging to directly observe the effect of MK-801 and sarcosine on neurons is a novel approach, and the findings support previous studies that state that sarcosine is a potential drug for the symptoms of schizophrenia [6,15,16,17,20]. Accumulating studies analyze calcium imaging in the hippocampus to display the cognitive function of animals [29,30,31,32]. However, some limitations or interesting findings were noticed during the experiments. We compared the advantages and caveats of the present methods with traditional unit recording here.

### 4.1. Fluorescence Intensity

The data demonstrated that regardless of the manipulations, the overall fluorescence intensity of trial B was very slightly lower than that of trial A (Figure 1I). Although the reductions in fluorescence intensity did not mask our main finding, which represented the significant decreases under the effect of MK-801 (Figure 1I), there is still some trade-off between long recording data and the photobleaching effect [23]. For a typical unit recording of place cells, a 10 min trial is common [33,34] or even up to 30 min [24]. However, we were concerned with photobleaching. Thus, we shortened the recording period to approximately 4 min per trial and 8 min per session. The problems of photobleaching could be improved by dimming the intensity of excitation light of the Miniscope [23] but may worsen the imaging quality if the signal–noise ratio of fluorescence is low. However, sarcosine still improved the fluorescence intensity of all mice (i.e., the red and blue lines shifted toward the right). While photobleaching could partially account for this reduction, we cannot rule out other factors, such as differences in neural activity or fluctuations in baseline fluorescence levels. However, we monitored the fluorescence signal over time and observed signs consistent with photobleaching, such as a gradual decline in intensity, which suggests that photobleaching is likely a contributing factor to the observed reduction.

The observation that low-dose sarcosine was more effective than high-dose administration (a similar trend is also observed in cell–cell correlation and spatial correlation analyses) could be related to the complexity of dose-response dynamics in NMDA receptor modulation. NMDA receptor is susceptible to desensitization, depending on the saturation of the glycine-binding site [35,36]. This desensitization happens because an overabundance of glycine weakens the glycine binding affinity on the NMDA receptors over time, causing the response of the NMDA receptors to fade [35]. A possible hypothesis for the opposing effects of a high dose of sarcosine is the endocytosis of NMDA receptors [36]. Consequently, a high dose of sarcosine may lead to a decrease in the overall NMDA receptor activity, reducing the therapeutic effects. We speculate that at lower doses, sarcosine may more selectively enhance NMDA receptor activity without overstimulating the system, thereby optimizing synaptic function. In contrast, higher doses may lead to receptor desensitization or saturation, reducing the therapeutic benefits. This observation suggests that careful dosing may be critical in therapeutic settings, warranting further investigation into the optimal dosing strategies for maximizing therapeutic effects. Research in human subjects has also shown that lower doses of sarcosine (1g/day) had a better impact on negative symptoms (measured by The Scale for the Assessment of Negative Symptoms (SANS)) compared to higher doses (2g/day) [15], which supports the effect observed in the present study. This parallel finding suggests that sarcosine’s efficacy might be optimized in certain tests at lower doses, potentially due to more targeted modulation of NMDA receptor function without overstimulation.

### 4.2. Measure the Grouped and Paired Cells

Figure 2 randomly sampled the cells and tested the fluorescence intensities with a unit of grouped cells. In addition, we further measured the correlation coefficients of cell–cell fluorescence intensities to test whether the cells fire together, which implies that they work together [37]. Much surprisingly, MK-801 dramatically increased the cell–cell correlations (Figure 3B,E). We postulate that the increased cell–cell correlation observed following MK801 administration may result from the widespread suppression of neuronal activity. MK801, by inhibiting NMDA receptor function, likely reduces the overall excitability of neurons, leading to a diminished fluorescence signal. This suppression may cause the neurons to enter a “synchronized” state of low activity, where they are not actively participating in distinct, independent network assemblies; the selective modulation of NMDA receptors by sarcosine may mitigate this abnormal synchrony. Instead of reflecting coordinated functional activity, this high correlation could be an artifact of reduced neuronal responsiveness, with most neurons displaying uniformly low levels of activity. Local field potential recordings have demonstrated that MK-801 induces hypersynchronization of gamma oscillations, indicative of aberrant network activity [38]. This dysregulated synchrony may contribute to a psychotic-like state, potentially mirroring the disrupted neural circuits characteristic of schizophrenia-related network dysfunction [38]. On the other hand, the increase in cell–cell correlation could also be explained by the impairment of organizing cell assemblies [39]. An increase in the coactivation of hippocampal neurons was reported in a tetrodotoxin-induced schizophrenia model that used electrophysiological recording [39]. Coactivation of cell assemblies that contain relevant functions or representations is one of the key mechanisms in the brain for processing information [37]. In contrast, suppressing unrelated cell assemblies is critical for organizing correct representations. Cognitive disorganization is one of the main symptoms that represent a deficit of inhibition of irrelevant mental activity in patients with schizophrenia [39,40]. We postulate that the increased cell–cell correlation after the injection of MK-801 may lead to the cognitive disorganization syndrome of schizophrenia and that sarcosine can improve disorganization in the hippocampus.

### 4.3. Physiological Function of dCA1 Neurons

Whether the function of dCA1 neurons is still intact after the administration of MK-801 is also an interesting question. Recording the properties of place cells in the dorsal hippocampus in a schizophrenia animal model has been reported [32,39]. However, various schizophrenia animal models combined with different recording tools revealed distinct results. Olypher et al. used tetrodotoxin to induce psychosis and applied a single-unit recording technique to acquire spikes from the dCA1 [39]. Their simulation model demonstrated that the ability to segregate relevant from irrelevant representations is impaired by tetrodotoxin [39]. Zaremba et al. used chronic two-photon Ca^2+^ imaging to track dCA1 place cell activities in Df(16)A+/− mice (an animal model of schizophrenia) [32]. They reported that Df(16)A+/− mice show a deficiency of stable spatial representation. Our data (Figure 4) have similar results to their report, which demonstrated diminished spatial correlation across trials under the influence of MK-801, and we also demonstrated that sarcosine significantly alleviated these decreases. However, we also noticed that the mean values of the correlation coefficient were low for all sessions (Figure 4B). We think the characteristics of calcium imaging and the size of the open field may account for the low spatial correlation; place cells were traditionally recorded by electrodes with a time resolution that is relatively more precise than calcium imaging for identifying the place field. We chose syn-jGcamp7s as a calcium indicator, given that it has a great signal–noise ratio [41]. However, the trade-off for a high signal/background ratio is time resolution. The half-decay time after 10 action potentials is approximately 1.26 ± 0.04 s [41]. Therefore, the time resolution of the calcium indicator is far from unit recording. In addition, because of the limitation of our recording equipment (i.e., lack of commutator) for recording on a large scale, we performed the experiment in a 30 × 15 cm open field. We postulated that a larger exploration arena for in vivo calcium imaging might reveal a place field map that is more identical to the traditional unit recording technique. Moreover, we did not specifically select and analyze cells that have place preferences. A place cell is easier to identify by repeatedly acquiring its place field in an identical environment. In Figure 4B, we assumed that all the cells we recorded were putative place cells, although the proportions of pyramidal cells that have place cell properties in the dCA1 are 60 to 70% [42]. The reason for making the assumption was due to the difficulty of performing multiple sessions a day without confounding photobleaching issues. Therefore, we did not design the protocol for three trials (i.e., baseline, MK-801, MK-801+SAR) session to compare their spatial correlations. Sarcosine may also improve synaptic plasticity in the dCA1 region [36] by reducing MK-801-induced hypersynchronization of neuronal activity, restoring normal network connectivity critical for coherent hippocampal place field properties. Place field impairments in MK-801-treated mice may be due to disruptions in NMDA receptor-dependent spatial encoding. The potential enhancement of spatial correlations by sarcosine may result from improved long-term potentiation [36], which is crucial for encoding hippocampal spatial representation and is facilitated by NMDA receptor activity.

### 4.4. Sarcosine Treatment

Abnormal dopamine or glutamate systems are two leading hypotheses of the pathophysiology of schizophrenia. In abnormal glutamate system theory, NMDA receptor hypofunction is a key cause of schizophrenia. The present study suggests that sarcosine may be beneficial to the NMDA receptor hypofunctional model of schizophrenia. The glycine modulatory site on the NR1 subunit of the NMDA receptor is a potential target for improving symptoms in schizophrenia [6,13,15,16,19,43], given that the concentration of glutamate in the central nervous system is difficult to elevate by peripheral administration. Therefore, an alternative strategy is used: elevating the production of glycine for the glycine modulatory site. In practice, the high effective dosage of direct administration of glycine is still a problem because of its poor ability to cross the blood–brain barrier, which results in a high effective dose and aversive effect for patients [14]. In contrast, inhibition of glycine transporter 1 is another potential way to treat schizophrenia. Sarcosine is a selective glycine transporter 1 inhibitor with a very wide margin of safety. Its recommended dosage is 2000 mg daily in humans, and the present study used a dosage of 500 mg/kg for mice, which is equivalent to 2349 mg for a 60 kg man [20]. Even 2000 mg or 4000 mg/day for humans reported no significant side effects [15,18,44]. Our data demonstrate that sarcosine administered at 500 mg/kg produced more consistent and pronounced improvements compared to the higher dose of 1000 mg/kg in our mouse model. By selectively inhibiting glycine transporter 1, sarcosine provides another therapeutic advantage and may reduce off-target effects in clinical use. Glycine transporters have distinct regional distributions: glycine transporter 1 is primarily located in the forebrain and hippocampus. In contrast, glycine transporter 2 is concentrated in the brainstem, cerebellum, and spinal cord [45]. Therefore, in our study, intraperitoneal sarcosine administration likely had minimal impact on glycinergic systems within the brainstem and spinal cord, thereby reducing potential effects on motor or sensory functions.

Moreover, sarcosine not only inhibits glycine transporter 1 but also regulates the surface trafficking of NMDA receptors, coactivates glycine modulatory sites, and enhances synaptic glycine levels [20]. Additionally, sarcosine may act as an agonist at the glycine modulatory site on NMDA receptors, potentially activating downstream BDNF/AKT/mTOR signaling pathways. This action could stabilize calcium influx and neural excitability [20]. Although accumulating reports show the pharmacological mechanisms of sarcosine and its effects during clinical use, the present study further provides some evidence for its improvement in patient hippocampal function.

### 4.5. Potential Sex Differences

This study exclusively used male mice as the animal model, which may introduce a potential bias toward findings relevant to males. Although the prevalence of schizophrenia does not significantly differ by sex [46], male patients tend to experience more severe negative symptoms [47]. Since NMDA receptor hypofunction is central to the pathophysiology of negative symptoms in schizophrenia [2,3,4,6,7], this study used sarcosine to ameliorate this dysfunction. However, NMDA receptor function may vary by sex. Studies indicate sex-dependent differences in NMDA receptor functioning in both physiological and pathological conditions [48,49]. For instance, female rats show lower NMDA receptor density in the CA1, CA2, and CA3 hippocampal regions during specific estrous stages [50] and exhibit higher expression of NR1 and NR2B subunits compared to males [51]. Moreover, estrogen may also have a protective effect on NMDA receptor function [52], potentially allowing for better resilience before the onset of symptoms. Consequently, we suspect that MK801 may exhibit greater potency in females due to its action on NR2 receptors [53], which are more prominently expressed in female rodents. However, the protective effect of estrogen should be considered. The sex differences in NMDA receptor function complicate predictions regarding the efficacy of sarcosine in female models. Further studies are warranted.

## 5. Conclusions

In summary, by directly and visually observing dCA1 neurons, our findings demonstrate that sarcosine mitigates the MK801-induced reduction in dCA1 neuronal activity and the associated increase in cell–cell correlation. In addition, combined with its underlying therapeutic mechanisms, a wide margin of safety, and influences on dCA1 neurons, sarcosine may ameliorate the hippocampal deficits in schizophrenia patients.

## Figures and Tables

**Figure 1 brainsci-14-01150-f001:**
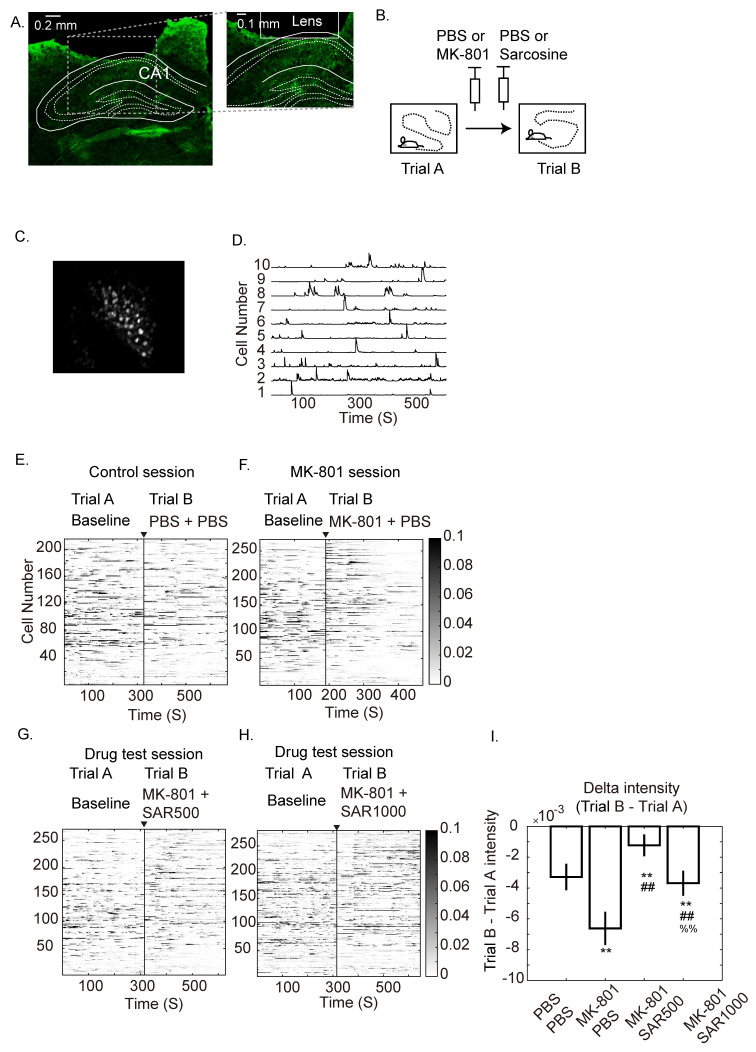
Examples of fluorescence intensity of dorsal CA1 neurons from mouse #3 after administration of MK-801 and sarcosine. (**A**) Histology slice for the location of GRIN lens implantation. (**B**) Experimental schedule of a session. The animal was explored in an identical open field in trials A and B. The drugs were injected before trial B. (**C**) Stacked image that was background-subtracted and movement-corrected (MC) by MIN1PIPE [26] of one recording session. The unit of the traces are normalized “processed calcium traces (PCT)”, which are normalized calcium signals after neural enhancement and denoising by MIN1PIPE (**D**) Examples of fluorescence traces. (**E**–**H**) Example of fluorescence intensities of the same animal under the treatment of PBS + PBS, MK-801 + PBS, MK-801 + sarcosine (SAR500, 500 mg/kg), and MK-801 + sarcosine (SAR1000, 1000 mg/kg). (**I**) Mean delta fluorescence intensities (trial B–A) of neurons of all animals (Mean ± SEM) (Each single cell was treated as a sample for bootstrapping before statistical analysis). **: vs. PBS + PBS; ##: vs MK-801 + PBS; %%: vs MK-801 + SAR500 indicates significant difference (*p* < 0.01), as determined using the Bootstrap method followed by Tukey’s post hoc test for multiple comparisons.

**Figure 2 brainsci-14-01150-f002:**
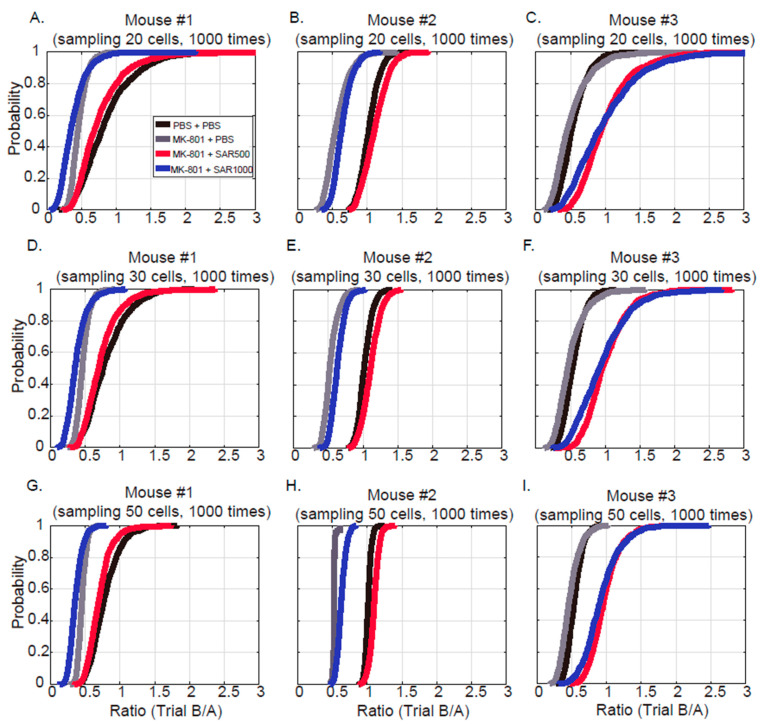
Fluorescence intensity ratio of trial B/trial A of grouped cell ensembles. (**A**–**C**) Cumulative distribution of the intensity ratio which is randomly sampled and averaged 20 intensity traces 1000 times in mice #1 to #3, respectively. The manipulations of PBS + PBS, MK-801 + PBS, MK-801 + SAR500, and MK-801 + SAR1000 are represented by black, gray, red, and blue lines. (**D**–**F**) Intensity ratio, which is randomly selected and averaged 30 intensity traces 1000 times. (**G**–**I**) Intensity ratio, which is randomly selected and averaged 50 intensity traces 1000 times.

**Figure 3 brainsci-14-01150-f003:**
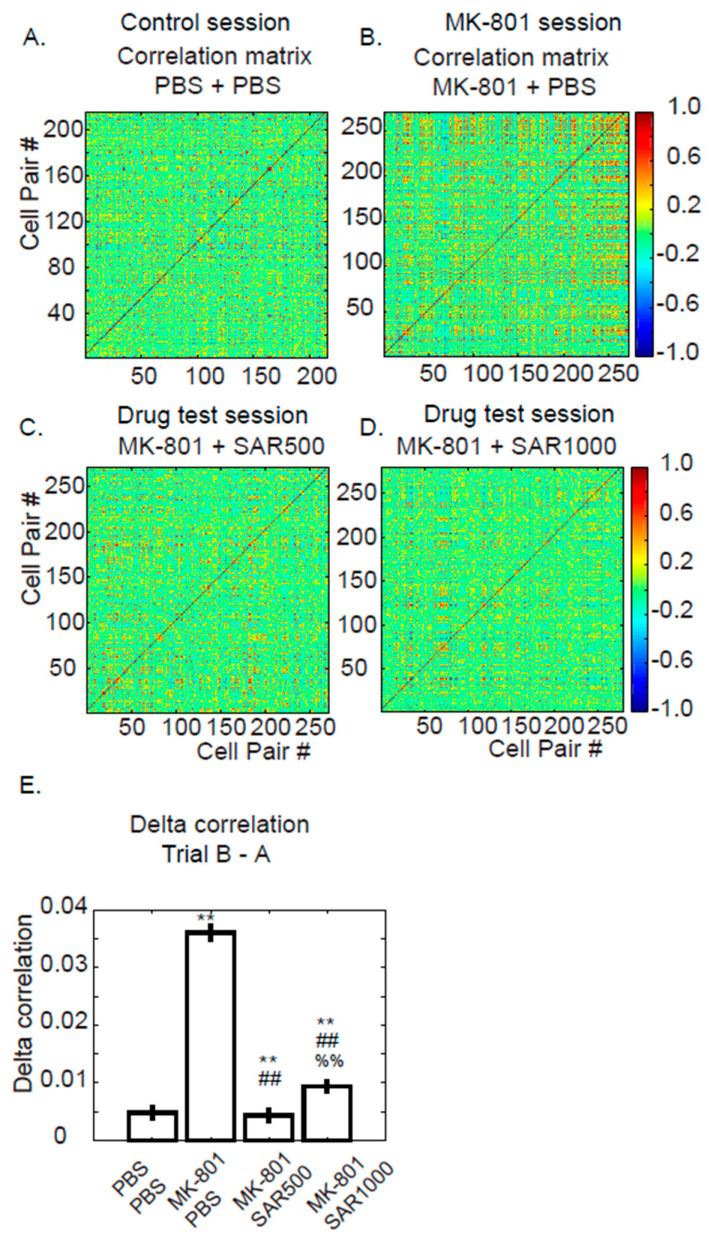
Examples of dorsal CA1 cell–cell correlation matrixes from mouse #3. (**A**–**D**) Cell–cell correlation matrixes after the administration of PBS + PBS, MK-801 + PBS, MK-801 + SAR500, and MK-801 + SAR1000, respectively. (**E**) Delta correlation coefficient (trial B–trial A) of pairwise neurons of all animals (Mean ± SEM) (Each single cell was treated as a sample for bootstrapping before statistical analysis). **: vs. PBS + PBS; ##: vs. MK-801 + PBS; %%: vs. MK-801 + SAR500 indicates significant difference (*p* < 0.01), as determined using the Bootstrap method followed by Tukey’s post hoc test for multiple comparisons.

**Figure 4 brainsci-14-01150-f004:**
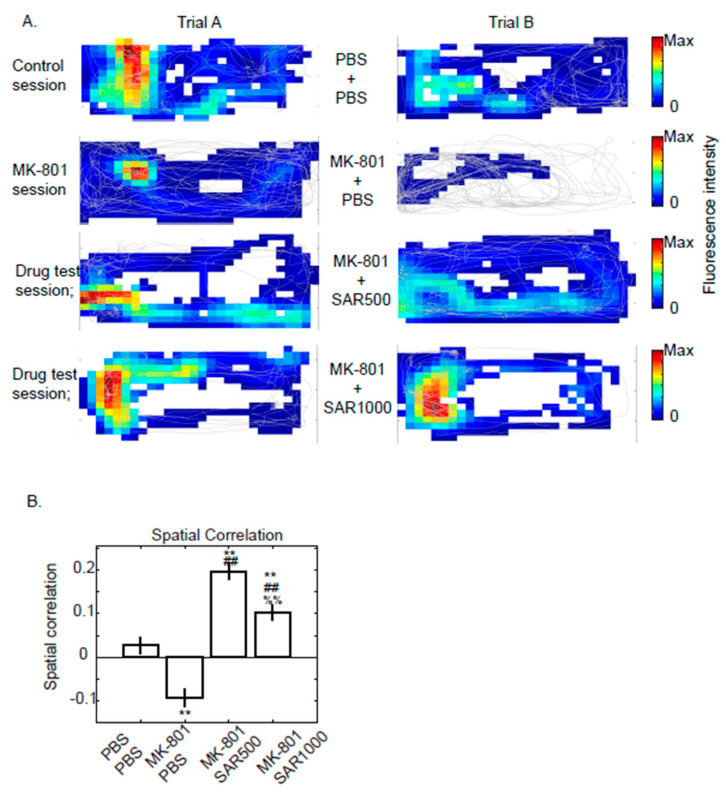
Examples of maximum fluorescence intensity maps from a fluorescence-positive cell in dorsal CA1. (**A**) Maximum fluorescence intensity maps of an example dorsal CA1 fluorescence positive cell are depicted by colors, where warm colors represent strong fluorescence signals and white represents weak signals. The gray line represents the walking trajectory of the animal (mouse #1). (**B**) Spatial correlation between trial A and trial B in neurons from all animals (Mean ± SEM) (Each single cell was treated as a sample for bootstrapping before statistical analysis). **: vs. PBS + PBS; ##: vs. MK-801 + PBS; %%: vs. MK-801 + SAR500 indicates significant difference (*p* < 0.01), as determined using the Bootstrap method followed by Tukey’s post hoc test for multiple comparisons.

**Table 1 brainsci-14-01150-t001:** The number of fluorescence-positive neurons for each mouse and treatment.

	PBS+PBS	MK801+PBS	MK801+SAR500	MK801+SAR1000
Mouse 10	199	101	150	161
Mouse 11	59	51	69	90
Mouse 12	215	271	270	279

## Data Availability

Data is available upon request by contacting the corresponding author due to the huge capacity of video data to upload.

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
