# Peer review of "Effects of Sarcosine (N-methylglycine) on NMDA (N-methyl-D-aspartate) Receptor Hypofunction Induced by MK801: In Vivo Calcium Imaging in the CA1 Region of the Dorsal Hippocampus"

_brainsci, 2024, doi:10.3390/brainsci14111150_

Round 1

Reviewer 1 Report

Comments and Suggestions for Authors

The study by Hsiao et al. reports the effects of sarcosine (n-methylglycine) on NMDA receptor  hypofunction induced by mk801. They used in vivo calcium imaging to observe the dynamics of fluorescence from the dCA1 hippocampal neurons. On the basis of the results obtained, the authors conclude that their data provide evidence supporting the use of sarcosine to alleviate symptoms of schizophrenia, especially hippocampus-related functions. The manuscript is well-written and well-organized. The results obtained are interesting although further investigations are needed. However, some points should be addressed before publication.

- The Authors wrote: “However, dopamine-targeting drugs do not alleviate both positive and negative symptoms”. This is not correct. Drugs targeting the dopaminergic system are effective for treating positive symptoms.

- There are typo throughout the manuscript. For example line 99: as pervious.

- The only limitation of this study is the exclusive use of male mice. In this respect, there are evidence for a sex-dependent functioning of NMDA receptors  in physiological an pathological conditions (PMID: 37293561; PMID: 32173404) This should be at least discussed.

- The discussion about potential mechanisms explaining the results obtained should be extended in order to improve the quality of the manuscripts.

Author Response

We are very grateful for the positive feedback and constructive suggestions. We have added several important references to strengthen the manuscript. With the Reviewers' guidance, we have made significant improvements to the quality of the current manuscript. The following is our point-by-point response to the Reviewer comments.

The study by Hsiao et al. reports the effects of sarcosine (n-methylglycine) on NMDA receptor hypofunction induced by mk801. They used in vivo calcium imaging to observe the dynamics of fluorescence from the dCA1 hippocampal neurons. On the basis of the results obtained, the authors conclude that their data provide evidence supporting the use of sarcosine to alleviate symptoms of schizophrenia, especially hippocampus-related functions. The manuscript is well-written and well-organized. The results obtained are interesting although further investigations are needed. However, some points should be addressed before publication.

- The Authors wrote: “However, dopamine-targeting drugs do not alleviate both positive and negative symptoms”. This is not correct. Drugs targeting the dopaminergic system are effective for treating positive symptoms.

Thank you for your feedback and for highlighting this clarification. We acknowledge that our original phrasing may have been misleading. Our intent was to indicate that dopamine-targeting drugs are primarily effective at alleviating positive symptoms, with limited impact on negative symptoms. We have revised the text accordingly to improve clarity. (see Line 48)

- There are typo throughout the manuscript. For example, line 99: as pervious.

Thank you for finding this typo for us. We have corrected it as suggested.

- The only limitation of this study is the exclusive use of male mice. In this respect, there are evidence for a sex-dependent functioning of NMDA receptors in physiological a pathological condition (PMID: 37293561; PMID: 32173404) This should be at least discussed.

We thank the Reviewer for the constructive suggestion, we have added section 4.5 (Line 526) to discuss the potential sex difference in the effects of sarcosine according the sex-dependent function of NMDA receptors.

- The discussion about potential mechanisms explaining the results obtained should be extended in order to improve the quality of the manuscripts.

Thank you for this valuable suggestion. We have added more detailed descriptions of potential mechanisms to the discussion section. In section 4.1 (Line 407-412) we extend the explanation of contrary effect of high-dose. In section 4.2 (Line 434), we emphasize sarcosine may mitigate abnormal synchrony. In Section 4.3 (Lines 485–492), we discuss the mechanisms by which sarcosine may enhance spatial correlation. In Section 4.4 (Line 519-525), we provide additional pharmacological explanations for sarcosine’s effects, expanding upon the previously mentioned mechanisms and including further potential pathways to clarify the observed results.

Reviewer 2 Report

Comments and Suggestions for Authors

The authors of “Effects of Sarcosine (N-methylglycine) on NMDA Receptor Hypofunction Induced by MK801: In Vivo Calcium Imaging in the CA1 Region of the Dorsal Hippocampus” explain their research on 3 mice implanted with miniscopes to visualize the neuronal activity on the dorsal CA1 neurons that previously were transfected with a fluorescent calcium marker. These mice were administered 4 different pharmacological treatments in a randomised way and studied their behaviour in an open field. The pharmacological treatments are based on the hypothesis that schizophrenia is the result of a hypofunction of the excitatory inputs. For this reason, they mimic this condition injecting MK801, that affects NMDA receptors making them less sensitive. To counteract this effect and try to explore its usefulness as a schizophrenia treatment, they also treat the animals with sarcosine (N-methylglycine), a glycine transporter-1 inhibitor at two different doses. The inhibition of the transporter provokes that glycine stays longer periods of time on the extracellular medium and can interact with the NMDA subunit, NR1, increasing the answer to the glutamate union to the receptor.

The paper has several limitations, starting with animal model is just a model based on a hypothesis that still have to be validated, the bleaching problem between sessions, or more pragmatic choices like the low number of animals, the time resolution of their calcium indicator or their open space characteristics, but, instead to hide them the authors are open to discuss about them showing their limitations and what kind of conclusions can still be extracted. So, I am quite satisfied of their paper, so my comments are mainly to clarify some points or discuss some points.

The authors should specify that sarcosine it is a specific inhibitor of GlyT1, but GlyT2 is not affected by it, So, its intraperitoneal injection has no effect on the Glycinergic inhibitory system of the brainstem and spinal cord.

On figure 1I, the y-axis legend says delta intensity, which is not quite understandable to the casual reader; probably it would be clearer “Trial B – Trial A intensity”

The section 3.2 Grouped fluorescence intensity, I think it should be explained in an easier or more descriptive way, since the concept averaging 20, 30 samples 1000 times it is not usual in the biological field. I know it is in other fields, and becoming more common in biology. However, some supplementary effort explaining it would be welcome for most of the readers. In addition, I kind of missing some hypothesis trying to explain why in two animals the sarcosine high dose had the contrary effect (moving it to the left instead of the right). Equally, their final phrase of the section should be narrowed down to low dose, since at 1000 their results does not support the affirmation.

Finally, the authors cite several time paper number 23 (Resendez et al), but probably some further explanation on the text would be welcome.

There is a typo on line 99, “pervious” instead of “previous”.

Author Response

We are very grateful for the positive feedback and constructive suggestions. We have added several important references to strengthen the manuscript. With the Reviewers' guidance, we have made significant improvements to the quality of the current manuscript. The following is our point-by-point response to the Reviewer comments.

Comments and Suggestions for Authors

The authors of “Effects of Sarcosine (N-methylglycine) on NMDA Receptor Hypofunction Induced by MK801: In Vivo Calcium Imaging in the CA1 Region of the Dorsal Hippocampus” explain their research on 3 mice implanted with miniscopes to visualize the neuronal activity on the dorsal CA1 neurons that previously were transfected with a fluorescent calcium marker. These mice were administered 4 different pharmacological treatments in a randomised way and studied their behaviour in an open field. The pharmacological treatments are based on the hypothesis that schizophrenia is the result of a hypofunction of the excitatory inputs. For this reason, they mimic this condition injecting MK801, that affects NMDA receptors making them less sensitive. To counteract this effect and try to explore its usefulness as a schizophrenia treatment, they also treat the animals with sarcosine (N-methylglycine), a glycine transporter-1 inhibitor at two different doses. The inhibition of the transporter provokes that glycine stays longer periods of time on the extracellular medium and can interact with the NMDA subunit, NR1, increasing the answer to the glutamate union to the receptor.

The paper has several limitations, starting with animal model is just a model based on a hypothesis that still have to be validated, the bleaching problem between sessions, or more pragmatic choices like the low number of animals, the time resolution of their calcium indicator or their open space characteristics, but, instead to hide them the authors are open to discuss about them showing their limitations and what kind of conclusions can still be extracted. So, I am quite satisfied of their paper, so my comments are mainly to clarify some points or discuss some points.

The authors should specify that sarcosine it is a specific inhibitor of GlyT1, but GlyT2 is not affected by it, So, its intraperitoneal injection has no effect on the Glycinergic inhibitory system of the brainstem and spinal cord.

We thank the Reviewer for the constructive suggestion, we have added the therapeutic advantage of sarcosine base on the distribution of GlyT (Section 4.4; Line 510-519).

On figure 1I, the y-axis legend says delta intensity, which is not quite understandable to the casual reader; probably it would be clearer “Trial B – Trial A intensity”

We have changed the label of y-axis to “Trial B – Trial A intensity” and upload the revised Fig.1

The section 3.2 Grouped fluorescence intensity, I think it should be explained in an easier or more descriptive way, since the concept averaging 20, 30 samples 1000 times it is not usual in the biological field. I know it is in other fields, and becoming more common in biology. However, some supplementary effort explaining it would be welcome for most of the readers. In addition, I kind of missing some hypothesis trying to explain why in two animals the sarcosine high dose had the contrary effect (moving it to the left instead of the right). Equally, their final phrase of the section should be narrowed down to low dose, since at 1000 their results does not support the affirmation.

We appreciate the Reviewer’s highlighting the clarity issues in our experimental description. We recognized that our initial explanation may have lacked detail. In response, we have now revised the Methods section (section 2.4.2 line 199 - 218) to provide a more descriptive and transparent account of our sampling approach, including the rationale behind this analytical method.

As the Reviewer’s suggestion regarding the explanation for the reduced efficacy at high doses of sarcosine, and we have clarified this in Section 4.1. Briefly, we propose that the effect arises due to high glycine levels, which may desensitize the NMDA receptor, reducing its responsiveness and thus weakening the improvement compared to low doses. Additional details are provided in Section 4.1 (Line 405-413) to enhance clarity on this mechanism.

Finally, the authors cite several time paper number 23 (Resendez et al), but probably some further explanation on the text would be welcome.

We referenced Resendez et al. (Paper 23) multiple times because it provides extensive guidance on techniques for in vivo head-mounted calcium imaging, including many critical protocols and techniques. We found it informative for developing our own protocol and included it for readers interested in technical details. We have also added a reference to a relevant video journal, offering a visual resource for readers seeking further practical insights into this methodology (line 123-124).

There is a typo on line 99, “pervious” instead of “previous”.

Thank you for finding this typo for us. We have corrected it as suggested.
